

# Influence of initial stratification, wind and sea ice on the modelled oceanic circulation in Nares Strait, northwest Greenland

Lovisa Waldrop Bergman[1] and Céline Heuzé[1,2]

[1]Department of Marine Sciences, University of Gothenburg, Box 461, 405 30, Gothenburg, Sweden
[2]Department of Earth Sciences, University of Gothenburg, Box 460, 405 30, Gothenburg, Sweden

**Correspondence:** Lovisa Waldrop Bergman (lovisawaldropbergman@gmail.com)

**Abstract.** Nares Strait in northwest Greenland is one of the main gateways for oceanic freshwater and heat exchanges between the Arctic and the North Atlantic. With a changing Arctic climate, understanding the processes that govern the oceanic circulation in Arctic straits has become crucial and urgent, but this cannot be done with current geographically and temporally sparse in-situ observations only. High resolution regional modelling is thus required, but costly. We here report on one-year sensitivity experiments performed with the coupled ice-ocean regional model MITgcm to determine the relative importance of wind forcing, initial stratification and sea ice thickness on the accuracy of the modelled oceanic circulation in Nares Strait. We find that the modelled basin's circulation is mainly driven by density gradients in the upper oceanic layer, making accurate initial fields of temperature and salinity essential for a realistic oceanic circulation. The influence of the wind and sea ice thickness is less important, potentially making such high resolution fields not necessary for accurate strait modelling, provided these results are valid for other sea ice models as well. Comparison with ship-based measurements collected in summer 2015 reveals the experiments to be too cold at the surface, probably because of a not-dynamic-enough sea ice cover. Although the modelled freshwater is rather accurate, large efforts need to be put into observing the ocean and the sources of freshwater continuously throughout the year to produce realistic and efficient model simulations of the Arctic Straits, key players in the entire Arctic system and global climate.

## 1 Introduction

The Arctic Ocean is changing in response to climate change, but when and how it will affect the global ocean circulation is still unclear. One crucial change is that in freshwater flux from the Arctic, caused by the melting of Greenland glaciers and an increase in sea ice export (Chen et al., 2006; Rignot et al., 2010). The freshwater flux from the Canadian archipelago (CAA) in particular, which accounts for more than a third of the total (Rabe et al., 2013), has increased by 50% from 40 to 60 mSv in the last two decades (Haine et al., 2015), which could have a weakening effect on the Atlantic Meridional Overturning Circulation (AMOC, Yang et al., 2016). The majority of the freshwater export via the CAA is through Nares Strait (Münchow, 2016),



between Ellesmere Island and Greenland (Figure 1). Nares Strait hence plays a crucial role in the Arctic freshwater budget and global climate system, and should ideally be accurately modelled for realistic present and future climate simulations.

But Nares Strait's dimensions make it challenging to model. The oceanic circulation in Nares Strait is characterised by a southward flow, where a subsurface jet with the width of the estimated Rossby radius of 5-10 km (Münchow et al., 2006, 2007)
dominates the southward volume and freshwater transports (Münchow et al., 2007). The total volume transport in the strait ranges from 0.71 to 1.03 Sv (Münchow, 2016), but these values are still uncertain due to the varying coverage by the moorings used to make the estimates and the short observation period (6 years). Moreover, Nares Strait is home to one of the largest marine-terminating glaciers of Greenland, Petermann Glacier, whose large basal melting can only be accurately modelled with a sub-kilometre grid resolution (Shroyer et al., 2017). But due to computational limitations, a resolution coarser than required
to even represent Nares Strait itself is often chosen to model pan-Arctic processes, which can cause a misrepresentation of the freshwater pathways in the Arctic (Lique et al., 2016). Regional configurations as used in this study may be the solution.

Previous regional modelling exercises in Nares Strait have shown a great sensitivity of the oceanic circulation to landfast ice cover, initial snow cover and atmospheric forcing (Shroyer et al., 2015, 2017; Castro-Morales et al., 2014; Koldunov et al., 2013). We here complement these studies by investigating the effect of extra initial conditions and forcings: the initial
ocean stratification, the spatio-temporal variability of the wind, and the initial sea ice thickness. We report on 12 one-year simulations performed with a regional setup of the numerical model MITgcm applied to Nares Strait. We investigate the resulting volume transport, current structure, the fresh water height, heat content and sea ice cover, and compare all simulations with observational data collected in Nares Strait in August 2015 (Heuzé et al., 2018). We conclude this work with a discussion on which simplifications can be made without drastically affecting the modelled circulation in the basin and which observation
strategy should be prioritised.

## 2 Model setup and experiments

To simulate the ocean circulation in Nares Strait we use a regional configuration of the general circulation model MITgcm (Marshall et al., 1997). The model domain ranges from 77.5°N to 82.0°N and 60.0°W to 78.5°W (Fig. 1, white box), giving an area of 255 km by 550 km. The horizontal grid resolution is 2.5 km in the across strait direction and 5 km in the along strait
direction, resulting in 102 by 110 grid cells. Due to local IT limitations, this is the highest resolution that can be achieved. The vertical grid has 50 layers with a minimum vertical spacing of 4 m towards the surface and increasing with depth to a maximum of 32 m. The simulation time is 274 days starting 1 January 2015 and finishing 2 October 2015, i.e. including the observation period of August 2015 (Heuzé et al., 2018). We applied a time step of six minutes for both the momentum equations and the tracer equations, which is the highest value achievable while still retaining numerical stability. The model gives an output every
two days.

We use the bathymetry from the International Bathymetric Chart of the Arctic Ocean (IBCAO) version 3.0 (Jakobsson et al., 2012), interpolated onto our model's higher resolution grid and smoothed with a moving average filter. Note that this bathymetry was created prior to the large calving events of Petermann Glacier in 2010 and 2012 (Falkner et al., 2011) and



hence is inaccurate in Petermann Fjord. The model domain has open boundaries with prescribed $u$ and $v$ velocities with a period of 1 month at the northern boundary extracted from the Max Planck Institute global coupled Earth System Model (MPI-ESM-MR, Giorgetta et al., 2013). To maintain mass balance in the basin, no velocities are prescribed at the southern boundary. A sponge layer is applied at both boundaries with a thickness of 10 grid cells, adding relaxation terms to the momentum and

tracer equations to smooth the transition between the prescribed velocities and the circulation further in the basin. A relaxation time of 12 hours was chosen for the cell closest to the boundary and a time of 10 days for the innermost sponge cell (M. Mazloff, personal communication November 2017). To represent the mixing processes in the ocean interior a nonlocal K-Profile Parametrization scheme (KPP, Large et al., 1994) is used. A no-slip condition is applied for the sides and the bottom of the basin.

We chose the dynamic and thermodynamic interactive sea-ice model from MITgcm (Losch et al., 2010) to represent the ice and its interaction with the ocean by directly affecting the heat fluxes, surface stresses and freshwater fluxes. The sea ice model uses the line successive over-relaxation technique (LSR) by Zhang and Hibler (1997) for the ice momentum equations and solves the equations for sea ice thickness, ice concentration, ice velocities and snow thickness. To simulate the import of ice in the northern part of the basin, a constant southward ice velocity of $10 \, \mathrm{cm\,s^{-1}}$ is prescribed at both boundaries. This

velocity is $15\text{-}25 \, \mathrm{cm\,s^{-1}}$ lower than what has been observed in Nares Strait (Ryan and Münchow, 2017), but tests with higher velocities resulted in unrealistic ice build up in the upper part of our domain. The imported sea ice has an concentration of 70%, which matches reasonably well with observed values, although the ice cover in Nares Strait is highly variable (Kwok, 2005). We applied a snow thickness of 10 cm, consistent with the observed mean value of 13 cm from 2005 (Haas et al., 2006), and we vary the initial sea ice thickness between 0 and 1 m. This is low compared to what has been estimated from satellite

remote sensing retrievals, especially when it comes to multi year ice that can reach a thickness of 4 m in the northern part of the Strait (Kwok et al., 2009), but again tests with higher values created unrealistic sea ice build up and caused the model to become computationally unstable. We hence chose to run with the lowest thicknesses within the observational range (Kwok et al., 2009).

To determine the model's sensitivity to initial conditions and forcings we use different input. However, all simulations

share some specific surface forcing due to the requirements of the coupled sea ice model. These consist of time-dependent surface forcing in the form of longwave downwards radiation, shortwave downwards radiation, 2 meter air temperature, specific humidity, sea surface temperature and precipitation from the ERA interim reanalysis produced by the European Centre for Medium-Range Weather Forecasts (ECMWF, Dee et al., 2011). All forcings retrieved from ECMWF have a period of 12 hours. The remaining forcings used in the simulations are wind speed, initial temperature and salinity fields, and sea ice thickness, for

which several simulations are conducted in order to determine their impact on the circulation in the basin. For the sensitivity experiments, the default settings are:

- Six-hourly wind fields from the HIRHAM regional climate model version 5 (Christensen et al., 2006) for the entire run period;

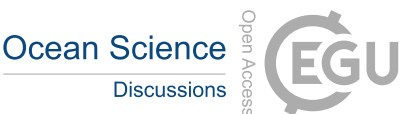

- Initial temperature and salinity fields from the Monthly Isopycnal and Mixed-layer Ocean Climatology (MIMOC, Schmidtko et al., 2013) for January;

- Initial sea ice concentration, sea ice thickness and snow depth from The Copernicus Marine Environment Monitoring Service (CMEMS, Von Schuckmann et al., 2016) for January 1st, 2015.

5 All forcing fields have been interpolated onto our domain's horizontal grid. The overall aim of this manuscript is to test the sensitivity of the model to different wind, initial stratification and sea ice thickness, as listed in table 1 and detailed below.

## 2.1 Wind experiments

To test the model's sensitivity to differences in wind forcing we conduct three experiments (Fig. 2):

- no wind;

10 - wind varying in space but constant in time;

- wind constant in space but varying in time.

For the case of spatially varying wind (Fig. 2b) we use a sinusoidal curve with a maximum value in the middle of the basin of -30 $\mathrm{m\,s}^{-1}$ and a minimum value at the boundaries of 0 $\mathrm{m\,s}^{-1}$. The wind is kept constant in the across-strait direction and in time. For the simulation with a temporally varying wind we use a similar sinusoidal shape, with its maximum value of -30 15 $\mathrm{m\,s}^{-1}$ at the start of simulation in January. The wind speed decreased to 0 $\mathrm{m\,s}^{-1}$ in June, and increased again in autumn (Fig. 2c). The wind speed for this simulation is kept spatially constant.

## 2.2 Stratification experiments

In order to determine the influence of the density gradients on the basin's circulation we conduct four experiments with different initial stratifications (Fig. 3):

20 - homogeneous 1-layer ocean;

- homogeneous 2-layer ocean;

- 2-layer with an East/West tilt;

- 2-layer with a South/North tilt.

For the 1-layer experiment, we impose a constant initial temperature $T = -0.5$°C and salinity $S = 33.5$ psu, which are the 25 average basin value in MIMOC. For the three 2-layer experiments, we use the average mixed layer depth (MLD) of 116 m from MIMOC as the boundary and the MIMOC mean T and S values above and below this depth. The upper layer hence has a temperature of $-1.4$°C and a salinity of 31.7 psu and the lower layer had a temperature of 0°C and a salinity of 34.4 psu. The two "tilted" stratification experiments (Table 1 and Fig. 3c and d) assume a tilt in the MLD from East to West and from South





### 2.3 Sea ice experiments

Our last four experiments address the influence of the initial ice cover on the oceanic circulation in Nares Strait (Table 1). In
the first sea ice experiment, we run MITgcm as ocean-only, i.e. with no sea ice model. In the second experiment, we include the
sea ice model but impose an initial sea ice cover and thickness of 0. For the last two experiments, the initial sea ice thickness
is 0.4 m and 1 m respectively; these two experiments share an initial ice concentration of 70% and an initial snow thickness of
10 cm. The same values are chosen for the imported ice at the boundary.

Since observations in Nares Strait are systematically restricted to the summer months and have large uncertainties attached
to them, the majority of our analyses consist in comparing the experiments among each other and to the literature to determine
which setting causes their differences. We also provide a brief comparison of our experiments with hydrographic observations
collected in Nares Strait in August 2015 (Heuzé et al., 2018) at the end of section 3. We quantify the effect of each experiment
on the ocean circulation, hydrography and sea ice by showing time-averaged maps of Nares Strait, time-averaged cross sections
(Fig. 1, chosen as the same section as the observations), and time series of the freshwater height, given by

$$F = \int_{h_2}^{h_1} \frac{S_{ref} - S(z)}{S_{ref}} \, dz, \tag{1}$$

and of the heat content

$$H = \rho c_p \int_{h_2}^{h_1} T(z) \, dz. \tag{2}$$

We here use $S_{ref} = 34.8\,\mathrm{psu}$, which is the maximum value of the salinity in the observational data; the mean density $\rho = 1028$
$\mathrm{kg\,m^{-3}}$, calculated using the reference salinity and a temperature of 0°C, which is close to the mean temperature in the basin;
and a heat capacity of $c_p = 3985\,\mathrm{J\,kg^{-1}K^{-1}}$.

### 3  Results and Discussion: across simulation differences

In this section, we first compare the experiments with each other to determine which setting has the largest impact on 1) the
ocean velocities, 2) the freshwater and 3) heat contents, and 4) the sea ice cover. We finish by evaluating our simulated results
against the observational data (5) described above.

### 3.1  Impact on the ocean circulation

The mean volume transport in the basin is determined by the boundary conditions set at initialization, and since all simulations
share the same prescribed velocities they all have the same resulting volume transport of 0.20 Sv, which is low compared to the





observational estimates of $0.71 - 1.03$ Sv (Münchow, 2016). Throughout the result section however, one should bear in mind that observations in Nares Strait are very few and have large uncertainties attached to them. It is hence not impossible that our modelled volume transport falls in the real observational range. The volume transport shows a predominantly southward transport for all experiments (Figure 4 blue colour), which is in agreement with previous modelling experiments in Nares Strait

(e.g. Shroyer et al., 2015).

The spatially and temporally varying winds (Fig. 4c and d) are the most different experiments, exhibiting an intense northward recirculation in the southern half of the domain. We suspect that this northward transport is a result of the balancing terms applied to the boundary, where the water is forced to recirculate north in order to retain mass balance in response to the strong winds. This means that applying a high velocity wind field in a basin with open boundaries may have a misleading effect on

the circulation due to the conditions set up at initialisation.The difference in volume transport between the default HIRHAM wind forcing and no wind forcing is surprisingly small (Fig. 4a and b). In contrast the difference between the default and the no wind simulations is one order of magnitude lower than for the other wind simulations, and gives rise to the question of whether it really is necessary to include a high resolution wind field when dealing with a basin partially covered with sea ice.

The different initial sea ice thicknesses also seem to have little impact on the overall volume transport in the basin (Figure

4, j-l). It can be that the differences tested here are too small to have an noticeable impact; previous modelling experiments in Nares Strait have shown that an sea ice thickness of 4 m drastically changes the circulation (Shroyer et al., 2015). Another possibility is that the Nares Strait circulation is mostly impacted by the formation and flushing of ice bridges (Shroyer et al., 2017), yet accurate formation of such ice bridges in models can only happen for a narrow range of external forcings and strait geometries (Rallabandi et al., 2017). But further studying these questions is however beyond the scope of this paper.

Finally, the vertical structure of the modelled ocean circulation (Fig. 5) is relatively accurate, exhibiting a two-cell structure with a northward transport towards the bottom and a southward jet 5-10 km wide Münchow et al. (2007). The jet is intensified by the surface in our experiments to a greater extent than has previously been observed, potentially because of our thin ice cover applied in these simulations. Shroyer et al. (2015) found that thicker ice results in a deepening of the southward jet. Reducing the initial stratification to homogeneous 1- and 2-layer systems removes the northward recirculation completely

(Fig. 5 e,f). The 2-layer with tilt experiments however do have the two cell structure (Fig. 5 g,h) indicating that the initial tilt in the stratification fields has a large impact on the resulting circulation in the basin. Accurate, relatively high resolution hydrographic observations are necessary for correctly initialising this tilt, and hence the entire strait circulation.

## 3.2   Impact on the basin-average freshwater content

To get a better understanding of how the basin reacts to the different forcings and the resulting differences in oceanic circulation

that we just highlighted, we now look at the time evolution of the freshwater height (FWH) averaged over the entire basin (Fig. 6). All experiments but the un-tilted stratification ones show an increase in freshwater from winter to summer, with a stabilisation around 12 m. The effect of the wind on the basin is most visible at the beginning of the simulation, where the spatially and temporally varying wind experiments have a lower FWH than the other simulations (Fig. 6a, February to April notably). This higher salinity could be a result of increased evaporation from the strong winds together with an increased export





of ice out of the basin (see appendix videos). When the temporally varying wind begins to subside the FWH increases, and the simulation moves towards the same values as the no wind case. Interestingly, the spatially varying (but constant in time) wind case is the one with the largest seasonal cycle, peaking in July before decreasing (Fig. 6a, green). This experiment is also the one with the largest ocean meridional velocities (Fig. 5c), which, when combined to the seasonally varying salinity, results in

the more obvious differences in freshwater than the other experiments.

The un-tilted experiments show hardly any time evolution of the freshwater content (Fig. 6b), staying at the low value at which they started. These experiments failed to create a realistic current structure (Fig. 5e and f) and hence were only driven by the prescribed velocity values at the boundaries and could not attain realistic hydrographic properties. With a tilt, the volume occupied by the top fresh layer increases compared to the simple 2-layer experiment (Fig. 6 purple and orange compared to

green), yet despite their start at different FWH they stabilise at the same value than the default experiment. In fact, of these three experiments the default one has the strongest surface current, followed by the E/W tilt, and the N/S tilt is the weakest (Fig. 5); the larger their surface current, the larger the increase in FWH throughout the run, potentially because the larger the sea ice mobility as will be discussed in two sections.

The sea ice simulations have similar FWH time series (Fig. 6). Surprisingly, removing the possibility for ice to form has a

very small impact on the salinity in our experiments, and towards the end of the simulation they all move towards the same value. This means that the circulation patterns might be more influential in the salinity change than the sea ice cover in the basin.

### 3.3 Impact on the basin-average heat content

Basin-average time series of the heat content (Fig. 7) show more differences between the experiments than the freshwater. The

temporally varying wind, 1-layer and no ice experiments exhibit a clear and similar seasonal cycle, with a decrease in heat in winter and increase in summer. Although seasonal observations in Nares Strait are lacking, such variations seem sensible. The spatially-varying experiment is not only the saltiest but also the coldest (Fig. 7a), suggesting a strong wind-driven evaporation throughout the run. The tilted stratification experiments in contrast are among the warmest (Fig. 7b), especially in summer, which combined with their realistic bottom intensified northward flow suggests a realistic northward import of comparatively

warm waters from Baffin Bay (Münchow et al., 2006).

The 1-layer stratification simulation shows a strong seasonal variation (Fig. 7b), with a large decrease in the beginning of the simulation, only to increase again in Mars to April. Due to the averaging of the MIMOC initial fields the surface water in the basin is warmer and saltier than in the other simulations, contributing to a thinner ice cover (see next section). This makes the water column more sensitive to the atmospheric heating and cooling, which is even more obvious for the no ice experiment

(Fig. 7c).

In summary, wind, stratification and sea ice experiments all seem to impact the modelled sea ice thickness, either via differences in sea ice export caused by the oceanic circulation or due to different surface and integrated hydrographic properties. We now quantify the actual differences in sea ice thickness throughout the run for all the experiments.



### 3.4 Impact on the sea ice thickness

Unlike the velocity, the average sea ice thickness presents a large spatial variability across the experiments (Fig. 8). The no wind experiment in particular that was extremely similar to the default runfor all our diagnostics, is now among the most different with very small thicknesses over the entire basin (Fig. 8b). It may be because this experiment is the warmest, with a consistently

high heat content throughout the entire run (Fig. 7a). The simplified varying wind experiments have some obvious issues as well (Fig. 8c and d). The spatially varying winds that are strongest at the center of the basin make the ice drift southward faster than it does by the southern boundary where the winds are weaker, which means that the ice builds up at the southern end of the domain. Despite the temporally varying case being relatively close to the observation-based climatology of Samelson and Barbour (2008), the sea ice in this experiment also builds up in the south, albeit to a lesser extent. A more realistic wind field

with a varying wind direction and synoptic events, like the one used for the default case, does not produce such an extreme situation and may be required for accurate modelling of Nares Strait and its mobile ice (Huntley and Ryan, 2018).

The surface water in the 1-layer case is initially warmer than in the other simulations due to the averaging of the MIMOC fields. It is therefore not surprising that the average ice thickness in the basin is low compared to the other experiments (Fig. 8e). Likewise, we just saw that the tilted experiments have a larger heat content than the default throughout most of the run

(Fig. 7), hence their time-average relatively low sea ice (Fig. 8g and h). But then, why isn't the experiment with the most heat of all, the 2-layer case, the one with the least ice? That is because heat is only one of the variables that control sea ice thickness. The time evolution of the basin-wide thickness (Fig. 9) shows that 2-layer, tilted N/S and tilted E/W coincide at the beginning of the run, but that from March onwards the 2-layer experiment has noticibly more ice than the tilted experiments. As shown on Figs 4 and 5, the tilted experiments have much stronger southward currents than the 2-layer experiment; they flush their sea

ice out of Nares Strait, when the 2-layer experiment only melts it locally, less fast (see the sea ice movies of all the experiments in the appendix).

All experiments exhibit the same seasonal cycle in sea ice, all having their maximum in April and minimum in August (Fig. 9), suggesting that the main control for the seasonal cycle lies in one of the forcing files that are common to all experiments, most likely the surface radiation. One exception is the temporally varying wind experiment (Fig. 9a), whose strong winds

in winter also flush the ice out of Nares Strait, resulting in a low winter thickness. Finally, although understanding the full dynamics of the sea ice model is beyond the scope of this paper, it is worth observing that differences in initial ice thickness persist throughout the run...except in the no initial ice experiment, which freezes up in less than a month and reaches the same thickness and coverage than the 0.4 m ice experiment (Fig. 9c). To the best of our knowledge, this model phenomenon has not been discussed in the literature. More experiments, initiated at different months of the year, would be required to understand

whether this is more than just a coincidence.

In summary, differences in the sea ice thickness in our experiments are caused by two processes:

– local melting / hindered sea ice formation, due to comparatively warm waters;

– and sea ice export out of Nares Strait, driven by strong winds and/or strong currents.



The currents themselves were shown in previous sections to be associated with the initial stratification and to impact the freshwater and heat content, whereas the wind and sea ice thickness experiments had little impact. In a final section, we investigate how realistic these results are when compared to observations.

### 3.5 Which is the most realistic simulation?

Most oceanographic measurements in Nares Strait have been collected over specific narrow sections in summer (see Johnson et al., 2011; Heuzé et al., 2017, and references therein). One of the main objectives of such regional modelling exercises as done in this manuscript is to expand these few observations, to the entire basin and/or to winter. We hence here conclude our study by comparing our twelve experiments to a synoptic section of observations collected across Nares Strait on 23 August 2015 (see Fig. 1).

The simulated freshwater content of the section is in good agreement with observations for the majority of experiments, except the 1- and 2-layer stratification ones (Table 2) that we have already identified as too salty on Fig. 6. All experiments have a similar heat content, only two thirds of that of the observations. Our experiments are missing heat at the surface: when the observations were collected, the area had been ice-free for several days and the top layer was up to $1°$ above freezing (Heuzé et al., 2017); in our experiments, it still is at freezing temperature. Looking at the sections themselves (Fig. 10) reveals

two different problems depending on the depth:

- the (southward) top layer is too salty in our experiments (red colour). It could be due to our too thin sea ice, i.e. that not enough sea ice volume is melted locally in our experiments; to issues at the boundary condition; or to the absence of surface and subglacial run off and glacial discharge from the neighbouring glaciers in our model, values that we chose to not try and include notably due to their large observational uncertainties (e.g. Rignot and Kanagaratnam, 2006; Rignot

and Steffen, 2008).

- the (northward) bottom layer is too fresh in our experiments (blue colour). The only experiment which is relatively accurate below 100 m is the 2-layer one (Fig. 10f), which is also the only one that has no northward flow (Fig. 4). This suggests that this experiment is initiated with a relatively accurate salinity which is then left undisturbed by the absence of northward currents, whereas in the other experiments what circulates (back) northward is too fresh. Modifying the

southern boundary conditions could improve this.

It is crucial to remember that the observational data only shows the salinity at one point in time and space, so the simulation results should not be dismissed based on these differences alone. More observations are required, over a longer time period, using new autonomous technology such as sea gliders and AUVs to observe year-round the entire water column even under sea ice (Webster et al., 2014; Johnsen et al., 2018). More modelling work is also required for the sea ice; we could not reproduce

the results of Rabe et al. (2012), who found that the geostrophic velocity in Nares Strait varies with the ice state and that the subsurface jet is displaced to the western side of the basin under fast ice conditions, simply because our model became unstable when we attempted to include thick sea ice. Moreover, the effect of Nares Strait's Petermann Glacier needs to be fully modelled, not only for its interaction with the sea ice cover (Shroyer et al., 2017), but also as surface and subglacial runoff



and calving, which all impact the freshwater content in Nares Strait (Münchow et al., 2006; Johnson et al., 2011; Heuzé et al., 2017). All these processes need to be estimated observationally as well. Only then can we hope to have realistic fully-coupled ice-ocean simulations of the freshwater system in Arctic Straits.

## 4    Conclusions

We conducted sensitivity experiments on a regional configuration of the ice-ocean model MITgcm in Nares Strait, northwest Greenland, to determine this region's oceanic circulation modelled response to different wind forcings, inital temperature and salinity fields, and initial sea ice thicknesses. High resolution wind from the state-of-the-art reanalysis HIRHAM and no wind at all resulted in a similar ocean circulation. Therefore, in ice-covered regions, working with low resolution wind fields that will save on computing resources may be good enough. Our results however disagree with those of Losch et al. (2010), who found that ocean models with thin sea ice react more quickly to more variable wind forcings; further study of how their sea ice model, which is the one coupled to the ocean here in MITgcm, handles the transfer of momentum from the wind into the ocean would be needed in order to explain this discrepancy. In agreement with Huntley and Ryan (2018), we found that the sea ice cover was affected by the strongest winds, either piling up unrealistically or flushing too fast out of the strait, and that such synoptic events need to be included in models. The main driver of the oceanic circulation in our experiments however was the density gradient, where the key component is the tilt of the upper layer. By using a simplified 2-layer system with a tilt from East to West or South to North, we recreate the two-cell circulation seen in the observations (e.g. Münchow et al., 2007). Finally, previous modelling experiments in Nares Strait have shown a big impact of the sea ice cover on the circulation when initialising with a thickness of 4 m (Shroyer et al., 2015); with our thinner ice, we found no such relationship. However, because of the large uncertainties connected with the estimations of the sea ice thickness (Kwok, 2005; Kwok et al., 2009), it is hard to determine which thickness is most representative of Nares Strait. This question is made even more complex by the presence of ice bridges, whose implementation in models is an active area of research (Rallabandi et al., 2017).

In conclusion, we argue that initialising the model with high resolution fields is not always necessary, provided the quality on the hydrographic observations is good. To date, this is not the case in Arctic straits, which are hard to reach and hence only observed for a few summer days every two-three years at most. Yet we showed that the initial tilt in stratification controls the oceanic circulation and resulting heat, freshwater and even sea ice in the strait; past studies have shown that it even controls the transfer of oceanic heat into the neighbouring marine-terminating Petermann Glacier (Johnson et al., 2011). Our results provide a first estimate of what would be an optimum observational strategy in Arctic straits, as well as in which simplifications can be done to the input datasets in order to reduce computational time, both points being necessary for a possible inclusion of Arctic straits in global coupled models and future accurate modelling of the Arctic freshwater budget.

*Code availability.*  The MITgcm source code is freely available at http://mitgcm.org/public/source_code.html





*Data availability.* CTD data from the Petermann15 expedition are freely available online at https://doi.pangaea.de/10.1594/PANGAEA.893180. Atmospheric forcing data from ECMWF are freely available at http://apps.ecmwf.int/datasets/data/interim-full-daily/levtype=sfc/. Temperature and salinity fields used for initialization are freely available at https://www.pmel.noaa.gov/mimoc/. Sea ice input data are freely

5    available at http://marine.copernicus.eu/. The velocity data used at the boundaries are freely available at https://www.mpimet.mpg.de/en/science/models/mpi-esm/. HIRHAM data can be freely downloaded from DMI e.g. at http://ensemblesrt3.dmi.dk/extended_table.html.

*Author contributions.* LWB and CH designed the experiments. LWB ran the simulations and analyzed the results. LWB and CH wrote the paper.

*Competing interests.* The authors declare no competing interest.

*Acknowledgements.* This work was supported by a VINNOVA Marie Curie Cofund fellowship awarded to C.H. (Dnr. 2015-01487). We

10   thank Ruth Mottram at DMI for providing the high resolution HIRHAM5 wind fields that were used in this work. We thank Sebastiaan Swart, Karen Assmann and Göran Broström for stimulating discussions during the project.



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



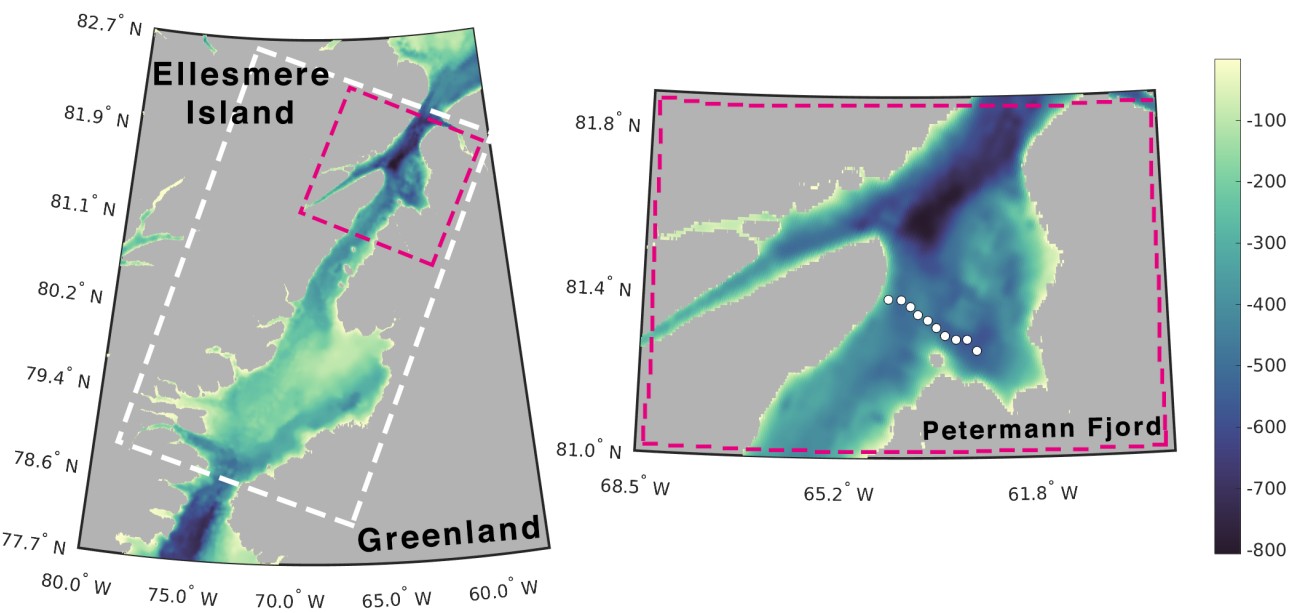

**Figure 1.** Left: Region bathymetry and model domain (white box). Right: Location of stations of August 2015 CTD casts used for validation in this study.





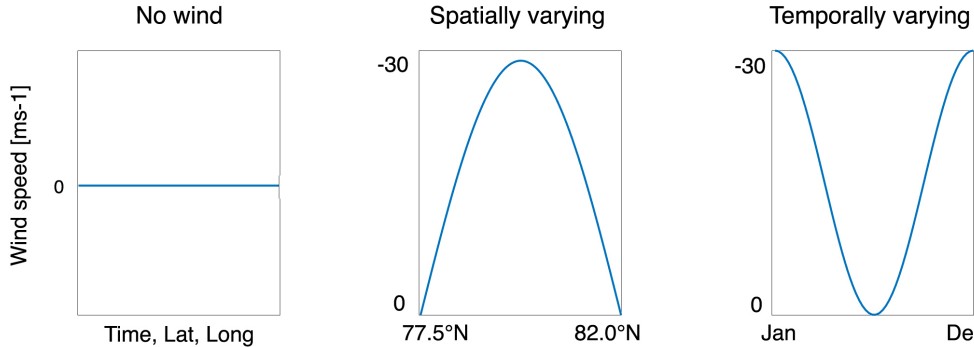

**Figure 2.** Wind forcing used for the simulations for a) No wind, b) Spatially varying wind and c) Temporally varying wind, all in ms$^{-1}$.





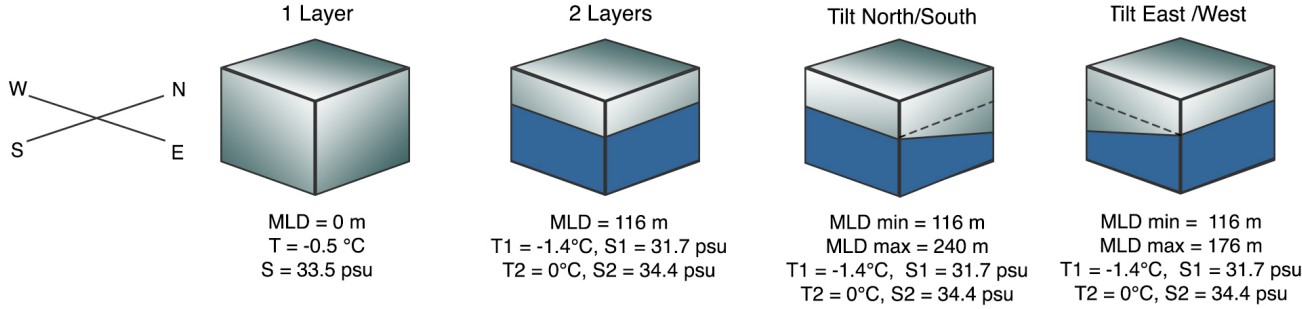

**Figure 3.** Initial stratifications used in the simulations with their respective mixed layer depth (MLD) for a) 1 layer, b) 2 layers, c) Tilt North/South and d) Tilt East/West with their respective temperature and salinity. Deepest MLD to the North and West respectively (see section 2). Index 1 refers to the top layer (grey); 2, to the bottom (blue).





**Figure 4.** Volume transport in Sverdrup ($10^6 \, \mathrm{m}^3 \, \mathrm{s}^{-1}$) averaged over the entire duration of the experiments. a) shows the default run, b) - d) the wind experiments, e) - h) the stratification experiments and i) - l) the sea ice experiments. Negative (blue) shading for southward transport.





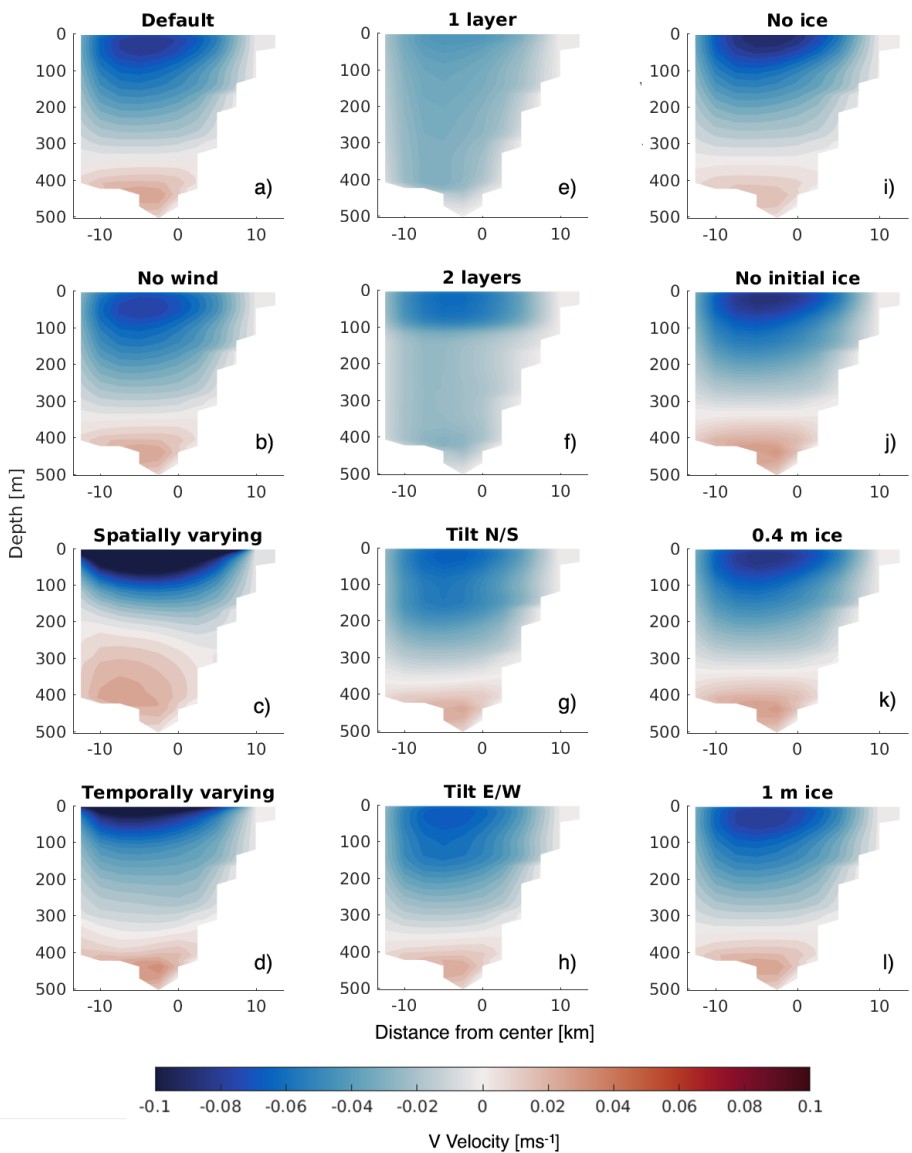

**Figure 5.** Cross section of the mean along strait velocity $(\mathrm{m\,s^{-1}})$ taken outside Petermann Fjord (see Fig. 1) averaged over the entire duration of the experiments; a) shows the default run, b) - d) the wind experiments, e) - h) the stratification experiments and i) - l) the sea ice experiments. Left is West; right, East; into the screen, northward (also red shading).




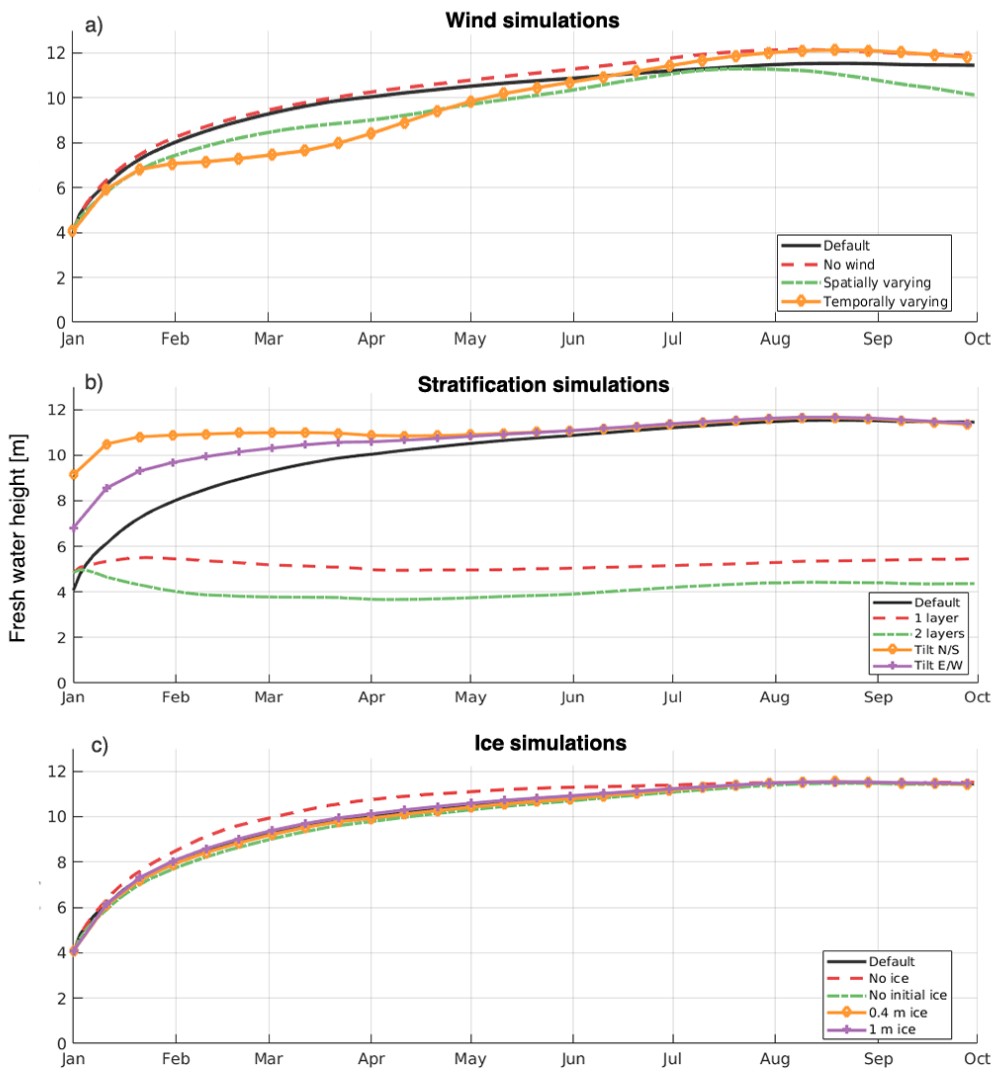

**Figure 6.** Time series of the basin-average freshwater content in meters. a) shows the wind experiments, b) the stratification experiments, and c) the sea ice experiments.



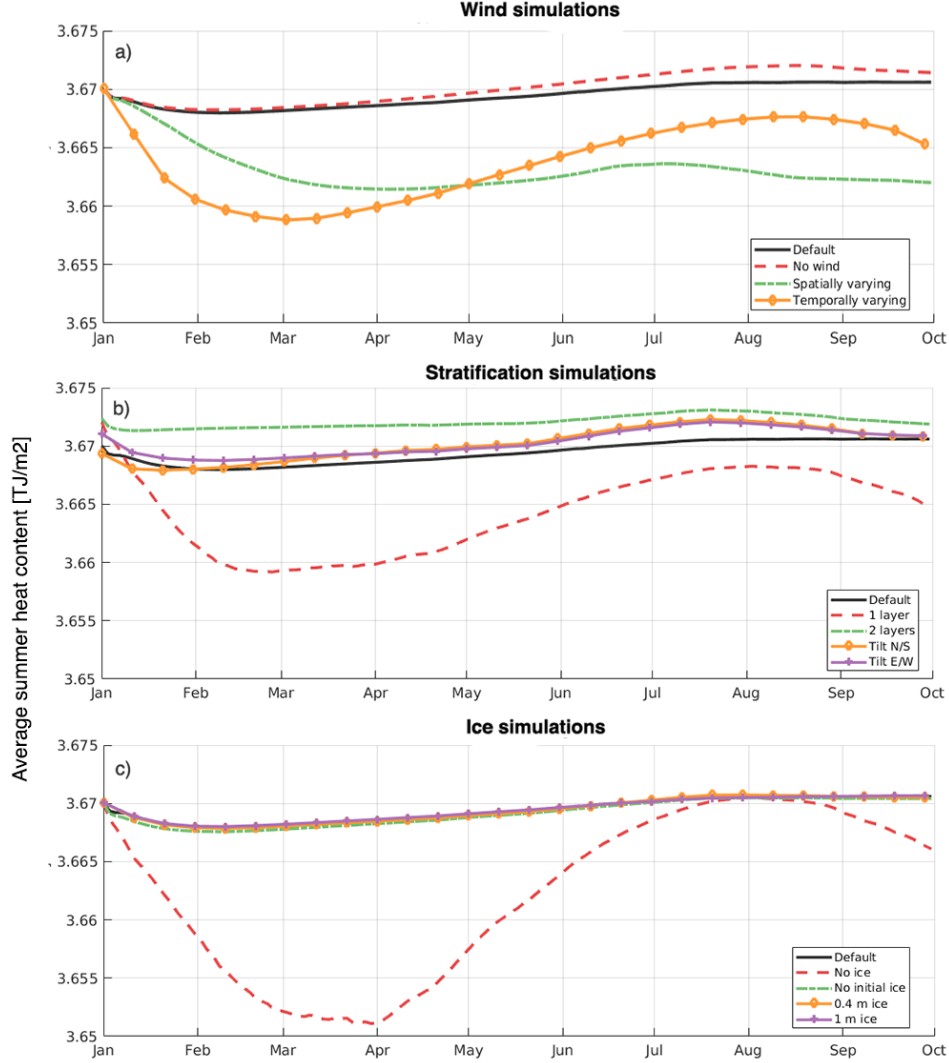

**Figure 7.** Time series of the basin-average heat content in TJ m$^{-2}$. a) shows the wind experiments, b) the stratification experiments, and c) the sea ice experiments.



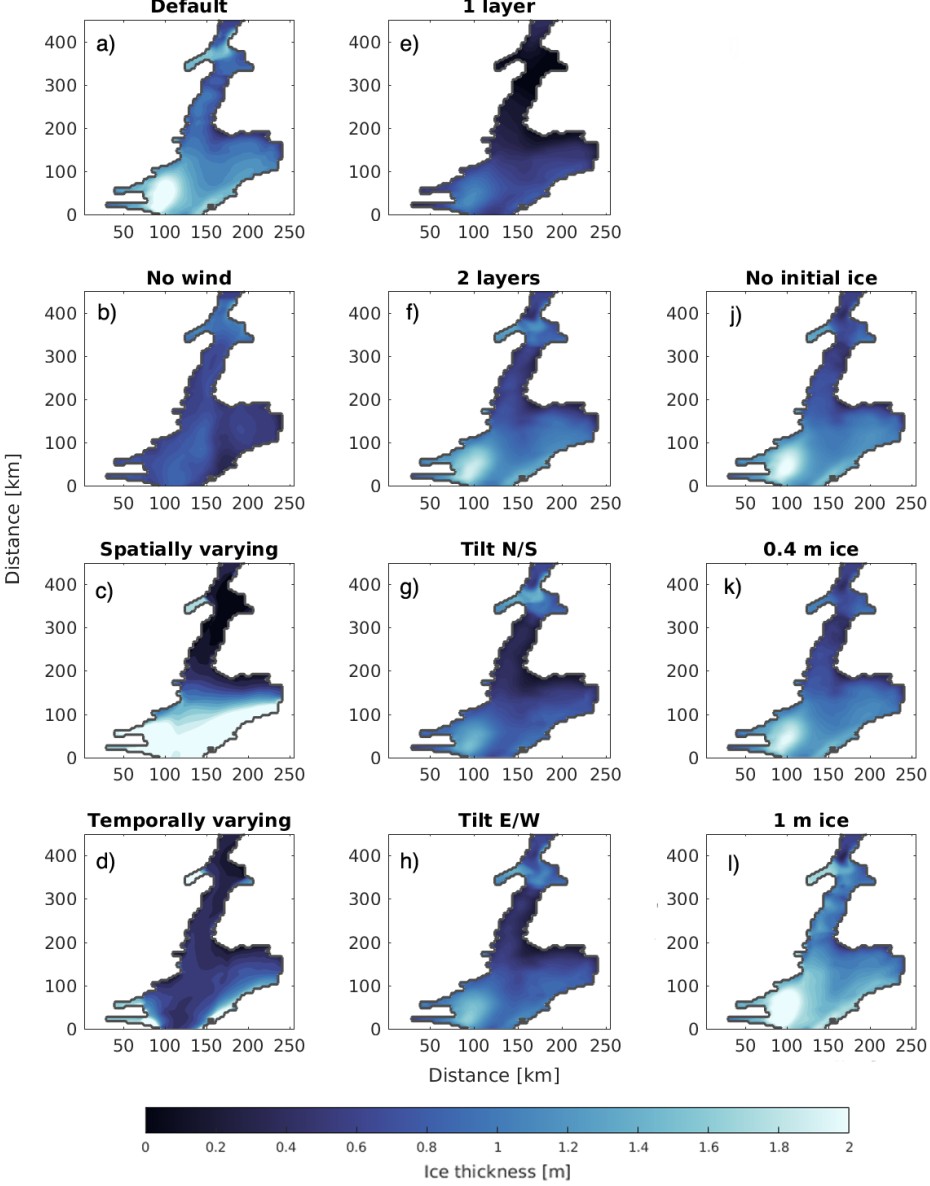

**Figure 8.** Sea ice thickness in metres, averaged over the entire duration of the experiments. a) shows the default run, b) - d) the wind experiments, e) - h) the stratification experiments and j) - l) the sea ice experiments.





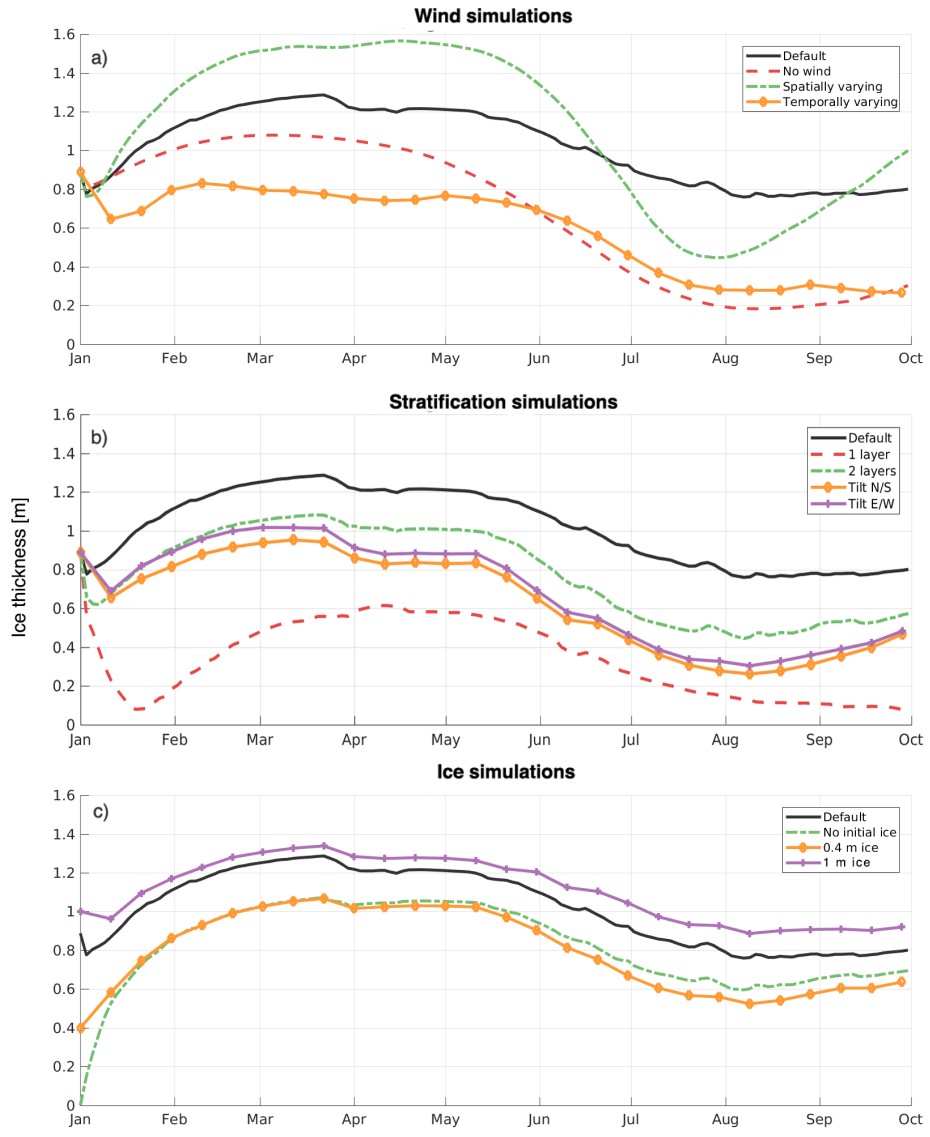

**Figure 9.** Time series of the basin-average sea ice thickness in metres. a) shows the wind experiments, b) the stratification experiments and c) the sea ice experiments.





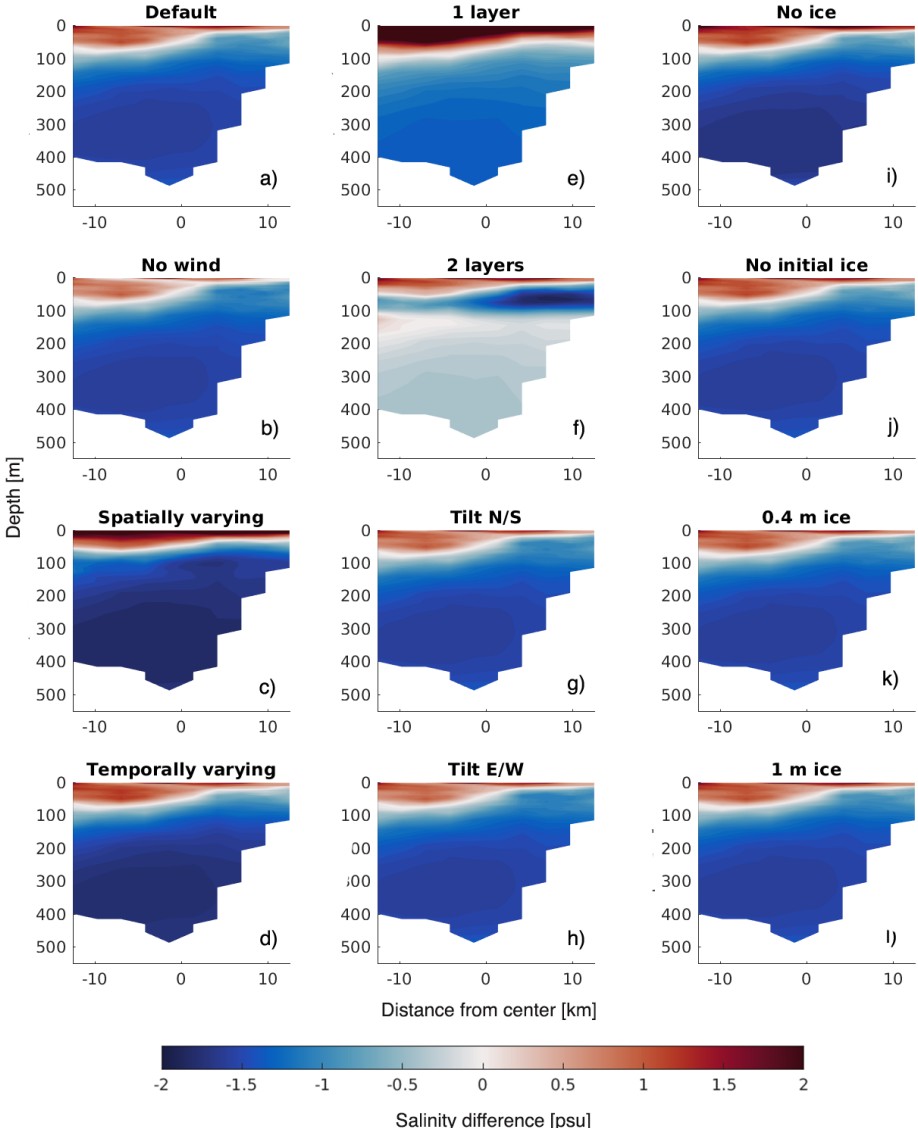

**Figure 10.** Salinity difference (experiment - observation) taken at the same date (23 August 2015) across the section highlighted on Fig. 1. a) shows the default run, b) - d) the wind experiments, e) - h) the stratification experiments and j) - l) the sea ice experiments. Left is West; right, East; into the screen, northward.



| Wind forcing | Initial stratification | Initial sea ice |
|---|---|---|
| HIRHAM *(default)* | MIMOC *(default)* | CMEMS *(default)* |
| No wind | 1-layer system | No sea ice model |
| Spatially varying | 2-layer system, uniform in space | Sea ice thickness = 0 m |
| Temporally varying | 2-layer with tilt South to North | Sea ice thickness = 0.4 m |
| | 2-layer with tilt East to West | Sea ice thickness = 1 m |

**Table 1.** The series of experiments conducted here. Default settings are indicated on the first line.





| Experiment | FWH [m] | Heat Content [TJ m$^{-2}$] |
|---|---|---|
| Observations | 11.91 | 6.1391 |
| Default | 11.09 | 3.9444 |
| No wind | 11.70 | 3.9446 |
| Spatially varying | 12.53 | 3.9339 |
| Temporally varying | 12.21 | 3.9380 |
| 1-layer | 5.80 | 3.9422 |
| 2-layers | 3.35 | 3.9454 |
| 2-layer with tilt N/S | 11.33 | 3.9450 |
| 2-layer with tilt E/W | 11.38 | 3.9448 |
| No sea ice model | 11.28 | 3.9456 |
| No initial sea ice | 11.00 | 3.9445 |
| 0.4 m sea ice | 11.12 | 3.9444 |
| 1 m sea ice | 11.10 | 3.9444 |

**Table 2.** Section-average freshwater height (FWH) and heat content in the observations (first line) and for our simulations, on 23 August 2015. Location highlighted on Fig. 1.