# Peer review of "Influence of initial stratification, wind and sea ice on the modelled oceanic circulation in Nares Strait, northwest Greenland"

_Ocean Science, 2018_

## Referee Comment (RC1) · Anonymous Referee #1 · 13 Dec 2018

This manuscript describes a superficial and failed ocean circulation modeling exercise for a region to the west of Greenland where thick multi-year sea ice is land-fast for most of the year in most years.

More successful, comprehensive, and credible models of ocean-ice interactions in Nares Strait already exist in the peer-reviewed literature, e.g., Shroyer et al. (2015, 2017) that are referenced by the authors and three more that are not, e.g.

1. Rasmussen et al. (2010, 2011) in Ocean Modeling and Atmosphere-Oceans from a

dissertation at Copenhagen University, e.g.

http://www.nbi.ku.dk/english/research/phd_theses/phd_theses_2010/till_rasmussen/Till_Anc

2. Wekerle et al. (2013) in J. Geophys. Res. from a dissertation at the University of Bremen, Germany, e.g.,

https://epic.awi.de/34179/1/Dissertation_Wekerle_Oct2013.pdf

3. Grivault et al (2018) in J. Geophys. Res. from a dissertation at the University of Alberta, Canada.

The present manuscript does not contain a single result or conclusion that is not already included in the above body of literature. Furthermore, the present work confuses many issues that are well observed and documented. Let me focus on just two major short-comings even though there are many more. These relate to ocean volume flux and sea ice implementations that perhaps explain why model results fail to relate to ocean or sea ice physics in Nares Strait:

The authors correctly observe that the volume flux through Nares Strait ranges from 0.71 to 1.03 Sv (interannual averages), but they discard these estimates as too uncertain, because of a short 6-year record. Yet, the numerical experiments are run for only 274 days long (page-2, line-27) and use a volume flux of 0.2 Sv (page-5, line-28) that is imposed via an upstream boundary condition. To me this contradiction explains why model results do not compare against from many years of mooring data. Failing to take this as an indication of model failure, the authors merely state that "we could not reproduce the results of Rabe et al. (2012) who found that the geostrophic velocity in Nares Strait varies with the ice state ... because our model became unstable when we attempted to include thick sea ice." (page-9, line-29)

This brings me the second major and fatal failure of this manuscript that relates to the sea ice as modeled and observed. It is well known and documented for 140 years that (a) Nares Strait is a conduit by which some of thickest Arctic sea ice leaves the

Arctic Ocean and (b) the sea ice cover is immobile for ~9 month a year in most years, see Kwok (2005, 2009) and Ryan and Muenchow (2017) for observational details from decadal data sets on sea ice coverage, mobility, and thickness. And yet, the authors are forced to run their models with ice initially between 0 and 1 m thin, because "... tests with higher values created unrealistic sea ice built up and caused the model to become computationally unstable." (page-3, line-21). A similar argument may apply also to sea ice velocities (page-3, line-15) that are kept artificially low. As a reviewer, I conclude that the sea ice model does not work in the present implementation. Prior ice-ocean models of Nares Strait listed above did not have this problem and compare more favorable to observations both qualitatively and quantitatively. Perhaps the authors should consult with sea-ice modelers before they release and distribute potentially faulty model results.

I have many additional issues with this manuscript that all relate to a naive reading, discussion, and understanding of the published literature (modeling and observations). The manuscript speculates excessively without much insight in either ocean or ice physics to "explain" model failures that are not recognized as such. One example of such speculation relates to the freshwater discharge through Nares Strait. On page-1, line-4 a 50% increase from 40 to 60 mSv (40,000 to 60,000 mˆ3/s) for the entire Canadian Archipelago, published values and time series exist for Nares Strait from moored instrumentation which vary from 40 to 50 mSv on average. Basal melting below Petermann Glacier is mentioned (page-2, line-7) as an additional source of freshwater flux that is not included in any models, however, the contribution of this large outlet glacier is about 2 mSv which is negligible relative to the 40-60 mSv freshwater flux.

There are too many (11) model experiments without a single one properly explained at the level that allows proper review. It appears that the boundary conditions control everything, yet pressure is never mentioned even though it may impose a dominant forcing as it does in observations. How do you define a volume flux (mˆ3/s) at a point as done in Figure-4. Salinity errors shown in Figure-10 frequently reach 0.5 to 1.5

psu within the top 30-50 m of the water column in (almost) all model experiments. Is it possible that mixed layer observations of active ice melt in the summer of 2015 are represented poorly by the model? Does the model ever have time to adjust to a steady state? Why use static metrics such as freshwater and heat content when more meaningful dynamic metrics such as freshwater and heat flux are available from published mooring observations?

I recommend to reject this manuscript without possibility of a resubmission, because excessive model instabilities, inconsistencies, and errors are more likely to explain discrepancies between prior published model results and observations that new physics. The impact of this manuscript on the community of oceanographers is negative as it may set a new low bar on the smallest publishable unit.

---

## Short Comment (SC1) · 17 Dec 2018

We thank the anonymous reviewer for their brutally honest comments, from which it became obvious that the aim of our study was not at all clearly stated. We shall reply in a similarly honest manner.

The reviewer is entirely right that we here report on a failed experiment. We failed at quickly implementing a model so that we could do some more interesting ocean circulation study. The reason we failed at it was that there was a surprisingly large

amount of parameters to set for which no suitable observational data was available. We obviously first checked against the literature, but found no ad-hoc solution. Hence, we turned that master's project into a basic sensitivity study.

We eventually obtained results, but we made it clear since the beginning that we did not trust them owing to the sea ice component with which none of us had any experience. So we asked; we asked in our institution; we asked on the MITgcm forum; we asked our network; we asked at Polar2018 where this work was presented. We asked but we never obtained help.

This is why we decided to submit our results to this journal: to initiate a discussion, so that we could eventually get the help we have desperately been needing. This has unfortunately failed as well, as we instead received rather clear suggestions from this reviewer to just give up on this work rather than try and fix it.

Nevertheless, we thank the anonymous reviewer for the time they have spent reading our work and writing their comments. We hope some other student will find either our work or the comments helpful, at least as guidelines as what not to do.

C. Heuzé

---

## Referee Comment (RC2) · Anonymous Referee #2 · 20 Dec 2018

The comment was uploaded in the form of a supplement:
https://www.ocean-sci-discuss.net/os-2018-122/os-2018-122-RC2-supplement.pdf

---

## Author Comment (AC1) · 14 Jan 2019

We thank the reviewer for taking the time to read our manuscript and giving us extremely helpful comments.

Reading these comments we have to agree that the choices for the atmospheric forcing are rather odd, both in the high wind speed and the source of the forcing fields. We will strive to avoid such inconsistencies in the future.

We also agree that the manuscript is still too long, despite having been significantly

shortened in order to match the requirements of a "technical note". But as the reviewer points out, as a result some important information is now missing from the manuscript, including some information about the initialization. We agree that it makes the experiments hard to evaluate, and we apologize for the lack of clarity.

Finally, the reviewer raised the question of whether the resolution could have an impact on the width of the subsurface jet. This is entirely possible and ideally the simulations would have been run with higher spatial resolution.

We will here not address their other comments, but we thank the reviewer all the same for providing them.

Lovisa Waldrop Bergman, Céline Heuzé

―――――――――――――――――――――――

---

## Author Comment (AC2) · 14 Jan 2019

We thank the reviewer for their valuable suggestions and taking the time to review our manuscript. We here address their main comments only.

The low volume transport in the model is a result of the velocities prescribed at the boundaries, and in the future we will take greater care to see that these velocities correspond better to the observed values.

Also, as explained in the letter that was unfortunately visible only to the editor, we were

aware that our experiments were far from successful, and initiated this discussion so that we could get helpful advice. We did wonder in particular whether starting the model at the sea ice minimum would have been more realistic, so we are glad to read the reviewer's comments. We will bear it in mind, along with all the others you wrote, when preparing future model runs.

As all three reviewers suggested a rejection, we will not address the other comments from the reviewer, but we thank them kindly for providing them.

Lovisa Waldrop Bergman, Céline Heuzé
* * *